# Neutralizing Antibody Response of the Wild-Type/Omicron BA.1 Bivalent Vaccine as the Second Booster Dose against Omicron BA.2 and BA.5

Hitoshi Kawasuji,[a] Yoshitomo Morinaga,[b,c] Hideki Tani,[b,d] Yumiko Saga,[b,d] Hiroshi Yamada,[b] Yoshihiro Yoshida,[a] Yusuke Takegoshi,[a] Makito Kaneda,[a] Yushi Murai,[a] Kou Kimoto,[a] Akitoshi Ueno,[a] Yuki Miyajima,[a] Kentaro Nagaoka,[a] Chikako Ono,[e,f] Yoshiharu Matsuura,[e,f] Hideki Niimi,[c,g] Yoshihiro Yamamoto[a,c]

[a]Department of Clinical Infectious Diseases, Toyama University Graduate School of Medicine and Pharmaceutical Sciences, Toyama, Japan
[b]Department of Microbiology, Toyama University Graduate School of Medicine and Pharmaceutical Sciences, Toyama, Japan
[c]Clinical and Research Center for Infectious Diseases, Toyama University Hospital, Toyama, Japan
[d]Department of Virology, Toyama Institute of Health, Toyama, Japan
[e]Laboratory of Virus Control, Center for Infectious Disease Education and Research (CiDER), Osaka University, Osaka, Japan
[f]Laboratory of Virus Control, Research Institute for Microbial Diseases (RIMD), Osaka University, Osaka, Japan
[g]Department of Clinical Laboratory and Molecular Pathology, Toyama University Graduate School of Medicine and Pharmaceutical Sciences, Toyama, Japan

**ABSTRACT** In addition to the original monovalent vaccines available for SARS-CoV-2, bivalent vaccines covering wild-type (WT) and Omicron BA.1 are also available. However, there is a lack of real-world data on the immunogenicity of bivalent vaccines as second boosters against the dominant Omicron sublineages, including BA.2 and BA.5. Healthcare workers ($n = 565$) who received the first booster vaccination were followed for 2 weeks after the second booster dose of the monovalent mRNA-1273 (WT group, $n = 168$) and bivalent BNT162b2 (WT+BA.1 group, $n = 23$) vaccines. Participants with previous SARS-CoV-2 infections were excluded from the study. The anti-receptor binding domain (RBD) antibody levels after the second booster dose in the WT and WT+BA.1 group were similar (median [interquartile range], 26,262.0 [16,951.0 to 38,137.0] U/mL versus 24,840.0 [14,828.0 to 41,460.0] U/mL, respectively). Although the neutralization activities of the pooled sera were lower against BA.5 than against other variants in both groups, the activities against BA.2 and BA.5 in the WT+BA.1 group were higher than those of the WT group in both pseudotyped and live virus assays. Vaccine-related symptoms, including systemic and local symptoms, were strongly correlated with anti-RBD antibody levels and neutralizing titers. In conclusion, the second booster dose of the bivalent (WT/Omicron BA.1) vaccine induced higher neutralizing activity against BA.2 and BA.5 than that of the original monovalent vaccine.

**IMPORTANCE** Although Omicron BA.1-containing bivalent vaccines have been authorized, real-world data validating their safety and antibody responses remain scarce. We conducted a prospective longitudinal study to assess the safety, immunogenicity, and reactogenicity of the second booster dose with the Omicron BA.1 bivalent vaccine in health care workers. Compared with the original monovalent vaccine, the bivalent (WT+BA.1) vaccine elicited higher levels of neutralizing antibodies against the Omicron BA.2 and BA.5 subvariants. The frequency of adverse events after the second booster dose was similar to that of the monovalent vaccine. BA.5-neutralizing antibodies induced by the bivalent Omicron BA.1-containing vaccine were expected to decline. A prospective longitudinal study should be performed to determine the persistence of the humoral immunity.

**KEYWORDS** BA.1, BA.5, Omicron, bivalent, neutralizing antibodies, secondary booster

Address correspondence to Yoshitomo Morinaga, morinaga@med.u-toyama.ac.jp.

The authors declare no conflict of interest.

The emergence of the Omicron lineage of SARS-CoV-2 was first recognized in November 2021, when it spread rapidly and became globally dominant (1). Omicron (B.1.1.529 [BA.1]) and Omicron subvariants (BA.2 and BA.5) are evolutionarily distant from the Wuhan variant (2). They have large numbers of substitutions in the spike protein that allow the virus to evade antibody neutralization and persistently be transmitted, decreasing vaccine efficacy (3, 4).

Vaccination programs against COVID-19 have been conducted worldwide; however, vaccine coverage varies per country based on the social circumstances of each country (5, 6). In May 2022, the Japanese government recommended a voluntary fourth dose to adults older than 60 years, immunocompromised individuals, and health care workers. A fourth dose of the original COVID-19 vaccine restores antibody levels; however, it provides only a modest short-term boost in protection against infection (7, 8). In particular, it is extremely difficult to induce immunity against BA.5 using the original vaccines (8).

SARS-CoV-2 continues to evolve, and the dominant variant keeps changing. Omicron BA.1 has been replaced by the BA.2 and BA.5 sublineages, whereas BQ.1 and XBB are already increasing in prevalence in some countries and regions (9). Bivalent vaccines are a strategy to protect against circulating variants and broaden neutralization to previous variants (10, 11). Interim data from phase 2 and 3 studies of bivalent vaccines covering the wild-type (WT) and Omicron BA.1, such as mRNA-1273.214 and Pfizer-BioNTech Bivalent, showed that they induced a higher humoral immunity against BA.2 and BA.5 than the original monovalent mRNA-1273 vaccine, which covers only the WT (11, 12). These bivalent vaccines have already been used, but not much information is available on their immunogenicity and safety compared to the monovalent vaccines.

It is crucial to assess newly authorized vaccines in real-world clinical settings and provide an accurate understanding of their humoral immune response to circulating variants and side effects. We conducted a prospective longitudinal study to assess the safety, immunogenicity, and reactogenicity of vaccination against SARS-CoV-2 in health care workers. In our facility, the original monovalent vaccine (mRNA-1273) or WT/Omicron BA.1 bivalent vaccine (BNT162b2) was administered as a second booster dose to health care workers. Our results revealed that the WT/BA.1 bivalent vaccine induced immunity against BA.2 and BA.5. Vaccine-related symptoms were similar to those of the first booster.

## RESULTS

**Study flow chart.** Anti-receptor binding domain (RBD) antibodies and neutralization activity against Omicron variant BA.1 were prevalent when the third dose of vaccination was promoted in Japan and were initially measured in 565, 425, and 321 participants at 2 weeks (2wA3D), 3 months (3mA3D), and 6 months (6mA3D) after the third dose (Fig. 1). To compare participants with a known vaccine history and known immune status, the humoral immunity of 203 participants was assessed at least once before the second booster dose. After the second booster vaccination, the participants who provided blood samples 2 weeks after the fourth dose (2wA4D) were divided into two groups: the original WT vaccine group (WT group) consisted of 168 participants who received the prototype mRNA-1273 vaccine as the second booster, and the other 23 participants received the recently authorized Pfizer BNT162b2 bivalent (WT/Omicron BA.1) vaccine.

**Antibody quantification and neutralizing activity before and after the second booster.** We previously reported the immunogenicity and safety of the BNT162b2 first booster vaccine at 2wA3D in the same participants as in the present study. We assessed the durability of the antibodies and their neutralizing activities after the first booster. At 3mA3D and 6mA3D, the median concentration of anti-RBD antibody was 8,955.0 U/mL (interquartile range [IQR], 5,879.0 to 13,612.0 U/mL) and 5,085.0 U/mL (IQR, 2,993.0 to 7,719.0 U/mL). The median high-throughput chemiluminescence reduction neutralization test (htCRNT) value for Omicron BA.1 was 85.8% (IQR, 75.2 to 91.6%) and 84.1% (IQR, 64.4 to 91.9%) (see Fig. S1A and B in the supplemental material).

As for the second booster vaccination, there were no significant differences in the demographic and immunological characteristics and humoral immunity levels, including

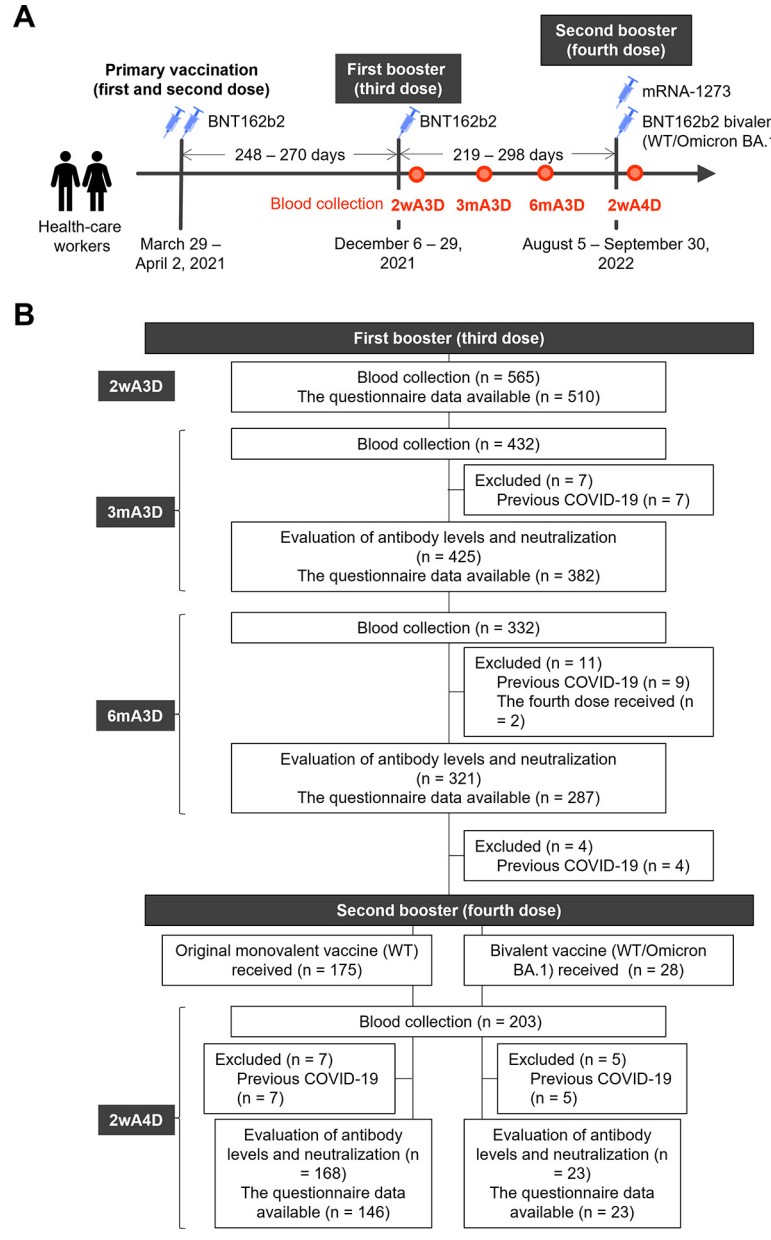

**FIG 1** Overview of the study. (A) A schematic illustration of the time course of the study. (B) Study flow chart. Participants ($n$ = 565) who received primary and the first booster vaccination of the BNT162b2 provided blood samples, and their vaccine-induced antibody responses were assessed at 2wA3D. They were subsequently followed up at 3mA3D and 6mA3D, and those with previous SARS-CoV-2 infection were excluded. A total of 203 participants who provided blood samples and whose antibody responses were assessed at least once after the third dose were eligible. The participants were divided into two groups after the second booster vaccination: the wild-type (WT) group consisting of 168 participants and the bivalent vaccine (WT+BA.1) group consisting of 23 participants. 2wA3D, 2 weeks after the third dose; 3mA3D, 3 months after the third dose; 6mA3D, 6 months after the third dose; 2wA4D, 2 weeks after the fourth dose; WT, wild type.

anti-RBD antibody levels and neutralization efficacy by htCRNT, between the WT and WT+BA.1 groups (Table 1).

At 2wA4D, the median concentration of anti-RBD antibodies was 26,262.0 U/mL (IQR, 16,951.0 to 38,137.0 U/mL) and 24,840.0 U/mL (IQR, 14,828.0 to 41,460.0 U/mL) in the WT and WT+BA.1 group, respectively (Fig. 2A). The htCRNT values, representing Omicron subvariant neutralization (Fig. 2B), using 100-fold sera against BA.2 in the WT+BA.1 group (median, >99.9% [IQR, 99.9 to >99.9%]) were significantly higher than those in the WT group (99.3%

**TABLE 1** Demographic and immunological characteristics of the study participants in the original mRNA-1273 group (wild-type [WT] group) and BNT162b2 bivalent (WT/Omicron BA.1) vaccine group (WT+BA.1 group)

| Profile | WT group, $n = 168$ | WT+BA.1 group, $n = 23$ | P value |
|---|---|---|---|
| Sex, male, n (%) | 40 (23.8) | 6 (26.1) | 0.80 |
| | | | |
| Age, yrs, n (%) | | | |
| 20–24 | 13 (7.7) | 2 (8.7) | 0.70 |
| 25–29 | 15 (8.9) | 3 (13.0) | 0.46 |
| 30–34 | 19 (11.3) | 5 (21.7) | 0.18 |
| 35–39 | 17 (10.1) | 2 (8.7) | >0.99 |
| 40–44 | 27 (16.1) | 2 (8.7) | 0.54 |
| 45–49 | 24 (14.3) | 4 (17.4) | 0.75 |
| 50–54 | 18 (10.7) | 1 (4.4) | 0.48 |
| 55–59 | 20 (11.9) | 4 (17.4) | 0.50 |
| 60–64 | 14 (8.3) | 0 (0.0) | 0.23 |
| ≥65 | 1 (0.6) | 0 (0.0) | >0.99 |
| | | | |
| Before the second booster dose | | | |
| No. of participants evaluated, n (%) | | | |
| 2wA3D | 168 (100.0) | 23 (100.0) | >0.99 |
| 3mA3D | 150 (89.3) | 19 (82.6) | 0.31 |
| 6mA3D | 134 (79.8) | 18 (78.3) | 0.79 |
| Anti-RBD antibody levels (U/mL, median [IQR]) | | | |
| 2wA3D | 22,242.0 (16,634.0–>25,000.0) | 22,699.0 (12,934.0–>25,000.0) | 0.68 |
| 3mA3D | 8,573.0 (5,652–13,727.0) | 13,001.0 (3,647.0–18,220.0) | 0.40 |
| 6mA3D | 4,288.0 (2,900.0–8,866.0) | 7,206.0 (2,821.0–13,708.0) | 0.27 |
| Neutralizing activity against Omicron BA.1-derived variant using 100-fold diluted sera, %, median (IQR) | | | |
| 2wA3D | 94.5 (92.4–96.5) | 93.6 (91.0–95.6) | 0.26 |
| 3mA3D | 85.4 (75.4–91.3) | 88.7 (76.1–94.0) | 0.18 |
| 6mA3D | 81.3 (61.2–91.2) | 83.1 (76.0–94.2) | 0.29 |

[IQR, 97.4 to 99.8%]). There was no difference against BA.5 (WT group, 99.3% [IQR, 98.4 to 99.6%] versus the WT+BA.1 group; 99.5% [IQR, 97.7 to 99.9%]). However, significant differences were more clearly observed when using 1,600-fold diluted sera against BA.2 (WT group, 38.5% [IQR, 10.0 to 61.8%] versus the WT+BA.1 group; 63.4% [IQR, 31.4 to 75.6%]) and BA.5 (WT group, 10.2% [IQR, 0 to 31.2%] versus the WT+BA.1 group; 22.7% [IQR, 14.3 to 41.0%]).

The pseudotyped virus-based half-maximal neutralizing titer ($NT_{50}$) values using pooled sera against WT, Omicron BA.1, BA.2, and BA.5 were ×400, ×400, ×100, and ×100 at

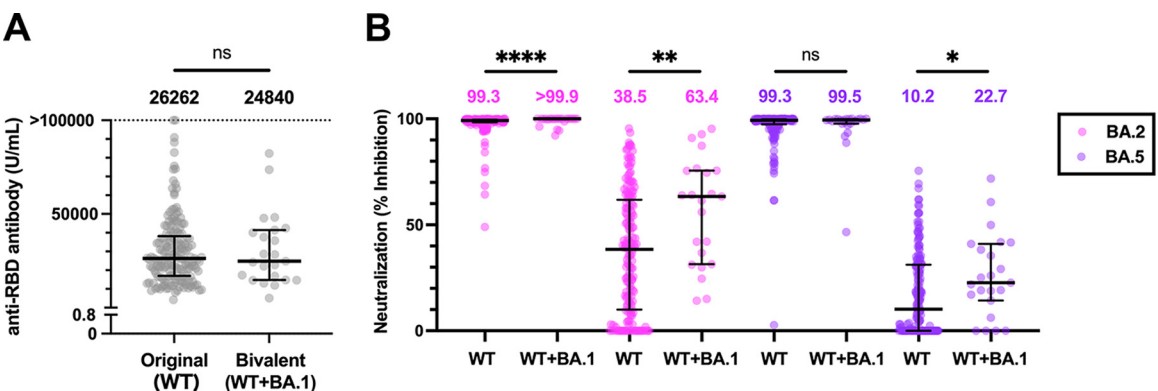

**FIG 2** Anti-RBD antibody levels and neutralizing activity after the second booster. (A) Serum concentration of anti-RBD antibody at 2wA4D in the WT group ($n = 168$) and the bivalent WT+BA.1 group ($n = 23$). Each dot represents an individual result. (B) Pseudotyped virus-based neutralizing activity against Omicron BA.2 and BA.5 at 2wA4D in the WT group ($n = 168$) and the WT+BA.1 group ($n = 23$). The assay was performed using 100- or 1,600-fold diluted serum. The numbers at the top indicate the median neutralizing values of each group. RBD, receptor-binding domain; 2wA4D, 2 weeks after the fourth dose; WT, wild type; *, $P < 0.05$; **, $P < 0.01$; ****, $P < 0.0001$; ns, not significant. Bars indicate medians with interquartile ranges.

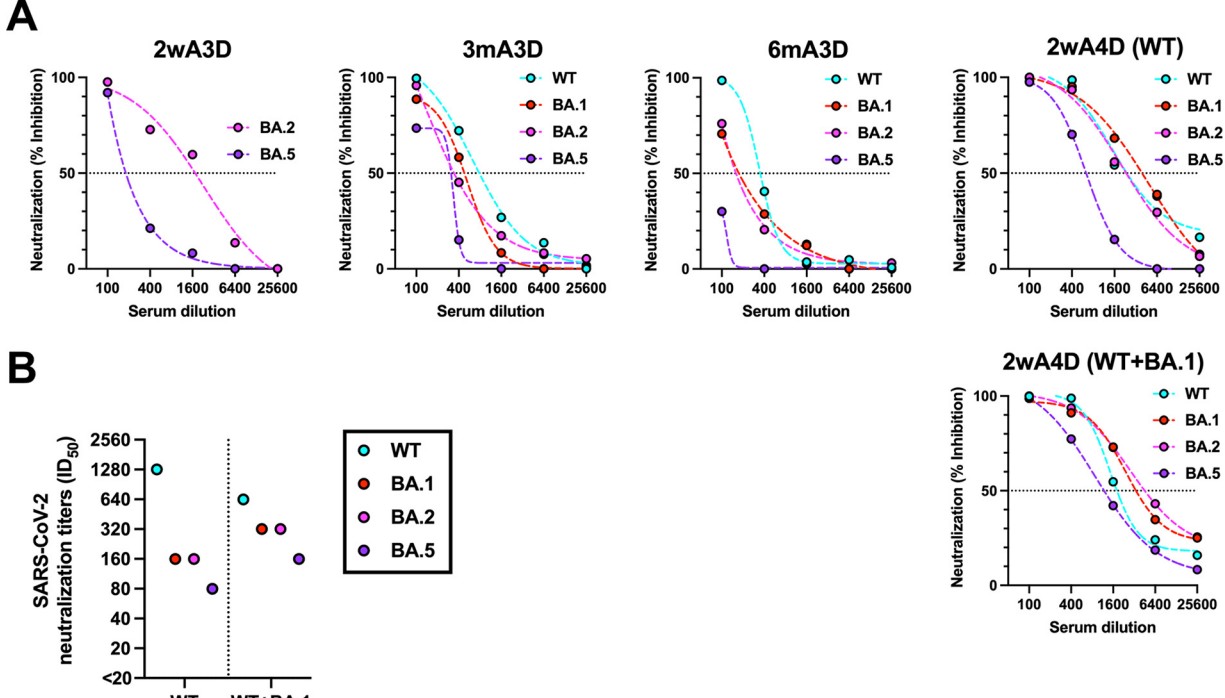

**FIG 3** Neutralizing activities before and after the second booster dose. (A) Neutralization titers ($NT_{50}$) against WT, Omicron BA.1, BA.2, and BA.5-pseudotyped viruses at 2wA3D ($n = 565$), 3mA3D ($n = 425$), 6mA3D ($n = 321$), and 2wA4D in the WT group ($n = 168$) and the WT+BA.1 group ($n = 23$) using the pooled serum. Dotted lines indicate interpolated standard curves. (B) The 50% inhibitory dilution ($ID_{50}$) titers against live viruses at 2wA4D in the WT ($n = 168$) group and in the WT+BA.1 group ($n = 23$). The assay was performed using pooled serum. WT, wild type; 2wA3D, 2 weeks after the third dose; 3mA3D, 3 months after the third dose; 6mA3D, 6 months after the third dose; 2wA4D, 2 weeks after the fourth dose.

3mA3D, and ×100, ×100, ×100, and <×100 at 6mA3D, respectively (Fig. 3A). At 2wA4D, neutralization was restored against all evaluated pseudotyped viruses in the WT and WT+BA.1 groups. The $NT_{50}$ values for WT, Omicron BA.1, BA.2, and BA.5 were 1,600, 1,600, 1,600, and 400, respectively. The serum of the WT+BA.1 group at 2wA4D showed higher $NT_{50}$ values against BA.5 than those of the WT group.

Live virus-based neutralization at 2wA4D also showed higher 50% inhibitory dilution ($ID_{50}$) titers against Omicron sublineage BA.1, BA.2, and BA.5 in the WT+BA.1 group than in the WT group. The $ID_{50}$ against the WT in the WT+BA.1 group was lower than that in the WT group (Fig. 3B). The second booster recovered the neutralization activity, which gradually decreased after the first booster (2wA3D, 3mA3D, and 6mA3D). The WT+BA.1 group showed higher $ID_{50}$ values against Omicron sublineages than those at 2wA3D (Fig. S2).

**Vaccine-related symptoms after the first and second booster.** Vaccine-related symptoms after the second booster dose were collected from participants in the WT and the WT+BA.1 groups (Table 2). Local and systemic symptoms were similarly observed in the WT and WT+BA.1 groups, with percentages of 94.5% and 84.3% in the WT group and 100% and 82.6% in the WT+BA.1 group, respectively. Frequent local reactions included pain at the injection site and local muscle pain. The most frequent systemic reactions were general fatigue, fever (temperature, ≥37.5°C), headache, and joint pain in both groups. Among the symptoms in the questionnaire, the incidence of diarrhea was significantly higher in the WT+BA.1 group (13.0%) than the WT group (2.1%).

Since immunogenicity after the first booster dose was maintained at a higher level in those who exhibited symptoms (Fig. S4A and S4B), the relationship between vaccine-related symptoms and humoral immunity after the second booster dose was investigated. At 2wA4D in the WT group, anti-RBD antibody levels were remarkably

**TABLE 2** Vaccine-related symptoms after the fourth dose of vaccination

| Symptom, n (%) | WT group, n = 146 | WT+BA.1 vaccine group, n = 23 | P value |
|---|---|---|---|
| Local symptoms | 138 (94.5) | 23 (100.0) | 0.60 |
| Pain at injection site | 109 (74.7) | 21 (91.3) | 0.11 |
| Redness | 25 (17.1) | 2 (8.7) | 0.54 |
| Swelling | 27 (18.5) | 1 (4.4) | 0.13 |
| Hardness | 11 (7.5) | 1 (4.4) | >0.99 |
| Local muscle pain | 68 (46.6) | 13 (56.5) | 0.50 |
| Feeling of warmth | 33 (22.6) | 3 (13.0) | 0.41 |
| Itching | 12 (8.2) | 0 (0.0) | 0.37 |
| Others | 3 (2.1) | 1 (4.4) | 0.45 |
| | | | |
| Systemic symptoms | 123 (84.3) | 19 (82.6) | 0.77 |
| Fever $\geq$ 37.5°C | 61 (41.8) | 9 (39.1) | >0.99 |
| General fatigue | 109 (74.7) | 16 (70.0) | 0.61 |
| Headache | 58 (39.7) | 7 (30.4) | 0.49 |
| Nasal discharge | 3 (2.1) | 1 (4.4) | 0.45 |
| Abdominal pain | 2 (1.4) | 2 (8.7) | 0.090 |
| Nausea | 13 (8.9) | 2 (8.7) | >0.99 |
| Diarrhea | 3 (2.1) | 3 (13.0) | 0.034 |
| Myalgia | 21 (14.4) | 1 (4.4) | 0.32 |
| Joint pain | 37 (25.3) | 5 (21.7) | 0.80 |
| Swelling of the lips and face | 0 (0.0) | 0 (0.0) | >0.99 |
| Hives | 0 (0.0) | 0 (0.0) | >0.99 |
| Cough | 1 (0.7) | 0 (0.0) | >0.99 |
| Others | 9 (6.2) | 3 (12.7) | 0.21 |

elevated in participants who presented with fever, general fatigue, and at least one systemic or local symptom (Fig. 4A). The antibodies neutralizing BA.2 and BA.5 were highly elevated in participants who had systemic symptoms (assay using 1,600-fold sera for BA.2 and assay using 100-fold sera for BA.5), fever and general fatigue, and had significantly higher neutralizing antibodies (Fig. 4B). In the WT+BA.1 group, no association between the symptoms and humoral immunity was observed.

The relationship between vaccine-related symptoms after the first and second booster doses was evaluated (Table S1). More than two-thirds of those who had experienced symptoms such as fever, general fatigue, or at least one systemic or local symptom after the first booster dose also displayed the same symptoms after the second booster dose. In contrast, symptoms absent after the first booster dose were observed in 27.6 to 71.4% in the WT group and 10.0 to 50.0% in the WT+BA.1 group.

## DISCUSSION

Although Omicron BA.1-containing bivalent vaccines have been authorized, real-world data validating their safety and antibody responses remain scarce. The C4591031 substudies D and E evaluated the safety, immunogenicity, and reactogenicity of the BNT162b2 bivalent (WT/Omicron BA.1) vaccine compared to the original BNT162b2 vaccine as a second booster dose in participants aged 55 and older that had not been previously infected with SARS-CoV-2 (13, 14). This clinical study showed that a second booster dose with the Omicron BA.1 bivalent vaccine elicited higher levels of neutralizing antibodies against BA.1, BA.4, BA.5, and BA.2.75 than the original monovalent vaccine (14).

However, data are still insufficient to ensure the safety and immunogenicity of the bivalent Omicron-containing vaccines. It is thus difficult for clinicians and policymakers to decide which booster to recommend while considering a delay of vaccination or vaccine shortages. Here, we report the first prospective longitudinal study to assess the safety, immunogenicity, and reactogenicity of a new BNT162b2 bivalent (WT/Omicron BA.1) vaccine in health care workers with no history of SARS-CoV-2 infection.

Similar to previous studies, we showed that the bivalent (WT+BA.1) vaccine elicited higher levels of neutralizing antibodies against Omicron BA.2 and BA.5 subvariants

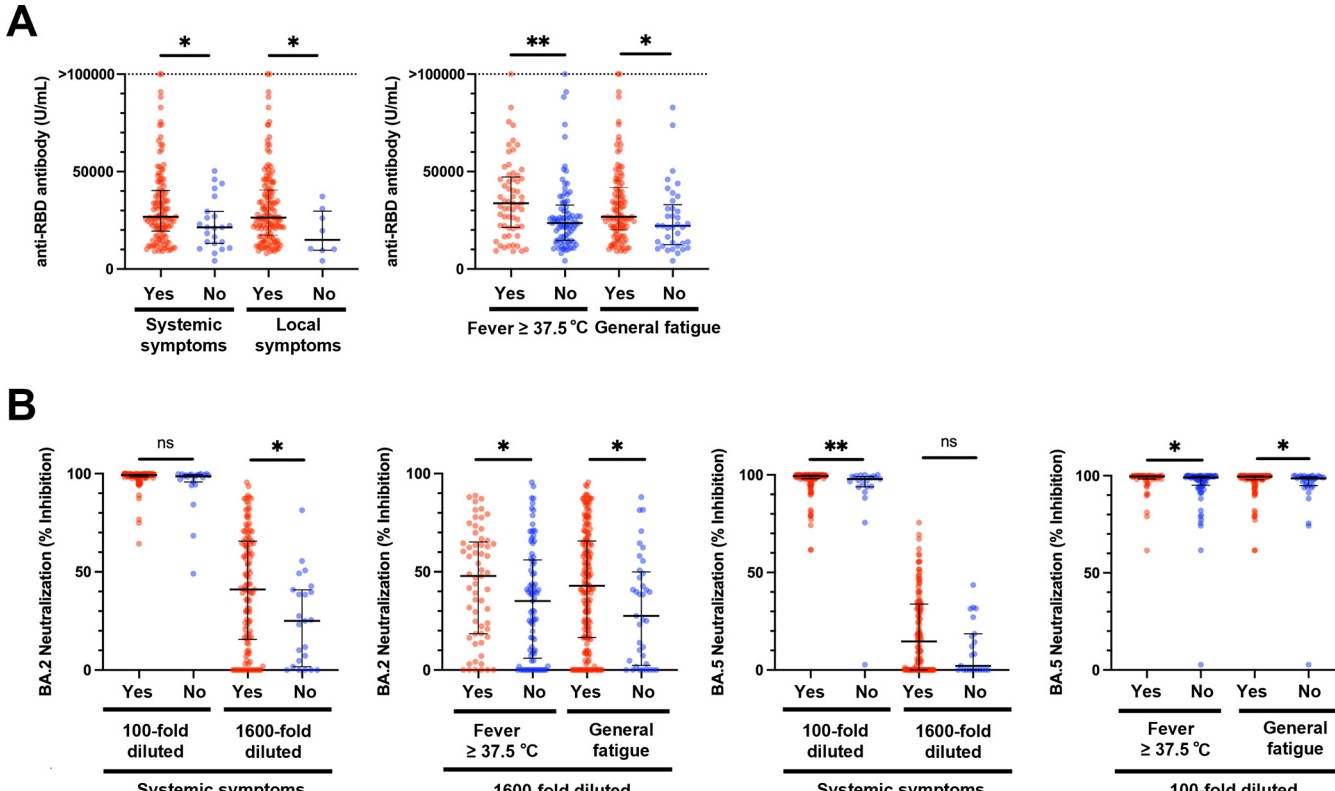

**FIG 4** Relationship of vaccine-induced antibody levels and vaccine-related symptoms after the second booster dose in the questionnaire-answered population. (A) Anti-RBD antibody levels in individuals with or without systemic or local symptoms (left) and specific symptoms (right) at 2wA4D in the WT group ($n = 146$). (B) Individual neutralizing activity against BA.2 (left panels) and BA.5 (right panels). RBD, receptor-binding domain; 2wA4D, 2 weeks after the fourth dose; WT, wild type. *, $P < 0.05$; **, $P < 0.01$; ns, not significant. Bars indicate medians with interquartile ranges.

than the monovalent vaccine (11). BA.5 evades other WT and Omicron BA.1 antibodies (15–17). Although BA.1-induced immunity is relatively less protective against BA.5, it provides some protection against reinfection with BA.2 and BA.5 (18, 19). Our data showed that the bivalent (WT+BA.1) vaccine produced 1.5 to 2.0-fold higher $NT_{50}$ and $ID_{50}$ values against BA.5 than monovalent vaccines that were still lower than those directed against other SARS-CoV-2 variants. This suggests that BA.5-neutralizing antibodies are induced by a single Omicron booster, although probably not sufficiently; the dosage therefore requires repeated boosters.

Following the approval of Omicron BA.1-containing bivalent vaccines, the current available Omicron BA.4/BA.5-adapted bivalent vaccines were rapidly authorized and are now widely used (12); however, the virus continues to evolve, and the BA.4 and BA.5 variants are no longer dominant in several countries (9). The convergent evolution of the BA.2, BA.4, and BA.5 lineages has led to the emergence of several new subvariants, including BQ.1.1, XBB.1.5, CH.1.1, and CA.3.1 (9, 20). Previous studies reported that a first or second booster dose of the bivalent BA.4/BA.5 vaccine was more immunogenic than the original monovalent vaccine against BQ.1.1, XBB, and XBB.1 (21, 22). However, recent studies have consistently demonstrated that the bivalent booster vaccines did not produce a robust neutralizing antibody response against BQ.1.1, XBB.1, XBB.1.5, CH.1.1, and CA.3.1 (20, 23). The newly emerged sublineages have accumulated additional spike mutations that enhanced immune evasion (24). It is therefore critical to monitor viral evolution and the impact on the immunogenicity and efficacy of the bivalent vaccines.

Long-term maintenance of vaccine-induced antibodies is important not only for an individual's protection against infection but also for social vaccination strategies. The comparison between the results of our previous report and our current study after the first booster dose led us to infer that the Omicron immunity after the original

monovalent vaccine rapidly decreases (25). This conforms with the results from similar studies (15, 26). In fact, vaccine effectiveness against infection waned remarkably within 3 months of administration in clinical settings (27, 28). Therefore, even when the second booster dose temporarily increase the antibody levels, they are expected to decline. In contrast, for the bivalent (WT/BA.1) vaccine, an ongoing prospective longitudinal study should be performed to determine the persistence of antibody levels after the second booster dose.

The evaluation of vaccine-related reactions is important for the continuation of repeated vaccinations. The frequency of adverse events after the second booster dose was similar to those after the first booster dose (11, 25). Similar to our previous studies about immunogenicity and safety following primary and booster vaccinations (25, 29), a positive relationship was observed between higher antibody responses and some specific adverse reactions. Furthermore, over 66.7% of participants who experienced symptoms after the first booster dose showed the same symptoms after the second booster dose. In contrast, those who showed symptoms after the first booster dose were less likely to experience the same adverse symptoms after the second booster dose. These findings suggest that acquiring immunity from vaccination is somewhat related to vaccine-related adverse events and that there are individual differences.

This study's chief limitation is that we included only participants aged 20 to 69 years, and simplified background information was collected. Thus, the antibody responses in older and younger individuals are unknown. In addition, since detailed interviews regarding specific underlying diseases and medications were not conducted, it was impossible to infer the impact of individual host health. Second, the number of participants in the BNT162b2 bivalent vaccine group was small because most health care workers (HCWs) had already received the second booster dose through vaccine campaigns before the bivalent vaccine was authorized. Third, the long-term persistence of neutralizing antibody levels after the second booster dose was not assessed, and neutralization activity against novel Omicron circulating variants (BQ.1.1, XBB.1.5, CH.1.1, and CA.3.1) was not measured. Fourth, the present study did not assess the vaccine efficacy and investigate the number of participants who became infected after receiving the second booster dose because the follow-up time after the second booster dose was limited. Future studies should focus on vaccine effectiveness and longitudinally monitor the incidences of symptomatic and asymptomatic SARS-CoV-2 infections in the WT and WT+BA.1 groups.

In conclusion, the second booster vaccination of the bivalent Omicron BA.1-containing BNT162b2 vaccine induced higher neutralizing activity against BA.2 and BA.5 than the prototype mRNA-1273 vaccine.

## MATERIALS AND METHODS

**Study design and participants.** We conducted this prospective longitudinal cohort study at Toyama University Hospital, a tertiary medical center in Japan with 612 beds and 1,639 health care workers. All participants were health care workers at this hospital and received the first booster (third dose) of BNT162b2 vaccine between 6 and 23 December 2021. This occurred an average of 260 (range, 248 to 270) days after the primary BNT162b2 vaccination (first and second doses). Participants were initially invited to provide blood samples 2wA3D, 3mA3D, and 6mA3D to assess humoral responses to the third vaccination dose. We previously reported the safety and immunogenicity at 2wA3D of the third BNT162b2 vaccine dose (25). In the present study, participants who provided blood samples at least once after the third dose were eligible for analysis and were voluntarily divided into two groups: those who received the original mRNA-1273 vaccine and those who received the recently authorized Pfizer BNT162b2 bivalent Omicron BA.1-containing vaccine as the second booster (fourth dose). The former received the original mRNA-1273 vaccine on 5 August, 10 August, and 2 September 2022; the latter received the BNT162b2 bivalent (WT/Omicron BA.1) vaccine on 30 September 2022. The second booster dose had been promoted in our hospital according to the national vaccine campaigns before the bivalent vaccine was authorized. Therefore, most health care workers (HCWs) had already received the second booster dose of the original monovalent vaccine, while those who had waited for the bivalent vaccines received the bivalent vaccine as a second booster dose later. Eligible participants were invited to participate in this study and provided peripheral blood samples 12 to 16 days after 2wA4D. Individuals with previous SARS-CoV-2 infections were excluded from the analysis. A previous infection was determined by a documented SARS-CoV-2-positive reverse transcription-PCR (RT-PCR) result or the presence of positive

anti-SARS-CoV-2 nucleocapsid antibodies. Anti-nucleocapsid antibodies in all participants were measured at 3mA3D, 6mA3D, and 2wA4D.

**Specimen collection.** Serum samples were collected from the participants at 2wA3D, 3mA3D, 6mA3D, and 2wA4D. The sera were used for serological assays within 3 days of storage at 4°C or frozen at −80°C until further verification.

**Outcome.** The primary outcome measures were differences in anti-RBD antibody levels and neutralization activity against the Omicron sublineage (BA.2 and BA.5) between the prototype mRNA-1273 and Pfizer BNT162b2 bivalent (WT/Omicron BA.1) vaccines at 2wA4D. We included subjects with a known vaccine history and immune status before the second booster dose. Secondary outcome measures were the incidence of local and systemic adverse effects after the mRNA-1273 or BNT162b2 bivalent vaccine dose and the association of antibody levels and neutralization activity with age, sex, and adverse effects in each vaccine. In addition, the association between adverse reactions after the third and fourth doses was also evaluated because identical individuals with information about adverse reactions after the third dose could be followed in this prospective longitudinal study.

**SARS-CoV-2 pseudotyped virus neutralization assay.** Pseudotyped vesicular stomatitis virus (VSVs) containing the SARS-CoV-2 S protein was generated as previously described (30). The expression plasmids for the truncated S protein of SARS-CoV-2 and pCAG-SARS-CoV-2 S (Wuhan) were provided by Shuetsu Fukushi of the National Institute of Infectious Diseases, Japan. pCAGG-pm3-SARS2-Shu-d19-B1.617.2 (Delta-derived variant), pCAGG-pm3-SARS2-Shu-d19-B1.1.529.1 (Omicron BA.1-derived variant), pCAGG-pm3-SARS2-Shu-d19-B1.1.529.2 (Omicron BA.2-derived variant), and pCAGG-pm3-SARS2-Shu-d19-B1.1.529.5 (Omicron BA.5-derived variant) were also generated. VSVs containing envelope (G) (VSV-G) were also generated. The pseudotyped VSVs were stored at −80°C until subsequent use.

The neutralizing effects of each sample against pseudotyped viruses were examined using an htCRNT as previously described (31). Briefly, serum was diluted 100 or 1,600-fold with Dulbecco's modified Eagle's medium (DMEM; Nacalai Tesque, Inc., Kyoto, Japan) containing 10% heat-inactivated fetal bovine serum and incubated with pseudotyped SARS-CoV-2 for 1 h. After incubation, the VeroE6/TMPRSS2 cells (JCRB1819) were treated with DMEM-containing serum and pseudotyped viruses. The infectivity of the pseudotyped viruses was determined by measuring the luciferase activity after 24 h of incubation at 37°C. The infectivity of samples without pseudotyped virus was defined as 0% infection, and that of pseudotyped virus without serum was defined as 100% infection (100% and 0% inhibition, respectively).

To measure the $NT_{50}$ values, the pooled samples were serially diluted by mixing equal volumes in a single tube, and neutralization activity was measured in duplicate by htCRNT. $NT_{50}$ was defined as the maximum serum dilution that indicated >50% inhibition.

**SARS-CoV-2 live virus neutralization assay.** SARS-CoV-2 isolates, WT (A, GISAID EPI ISL:408667), Alpha (B.1.1.7, GISAID EPI ISL:804007), Beta (B.1.351, GISAID EPI ISL:1123289), Gamma (P.1, GISAID EPI ISL:877769), Delta (B.1.617.2, GISAID EPI ISL:2158617), Omicron sublineage BA.1 (BA.1.1, GISAID EPI ISL:7571618), Omicron sublineage BA.2 (BA.2, GISAID EPI ISL:9595859), and Omicron sublineage BA.5 (BA.5, GISAID EPI ISL:13241867) were kindly provided by the National Institute of Infectious Diseases (Japan). VeroE6/TMPRSS2 cells were infected with SARS-CoV-2 isolates to obtain a high titer of the virus stock. The cell culture medium was harvested after 2 or 3 days of inoculation and centrifuged. The virus-containing supernatant was stored at −80°C. Prior to the neutralization experiments, viral titers were defined by median tissue culture infectious dose ($TCID_{50}$).

Using the same pooled samples as the pseudotyped virus assay, the serum infection-neutralization capacity was analyzed in duplicate by testing 2-fold serial dilutions of sera, starting at 1/20, with 50 $TCID_{50}$ of the virus in VeroE6/TMPRSS2 cells in 4 wells of 96-well plates under biosafety level 3 conditions. After 4 or 5 days of incubation, cells were fixed with paraformaldehyde and stained with an aqueous crystal violet methanol solution. The serum titer ($ID_{50}$) that showed 50% protection from virus-induced cytopathic effects was considered to contain neutralizing antibodies and was defined as the reciprocal value of the sample dilution. Each run included an uninfected cell control, an infected cell control, and virus back-titration to confirm the virus inoculum.

**Anti-RBD and anti-nucleocapsid antibody measurements.** The concentration of anti-RBD antibodies in serum samples was measured using the Elecsys anti-SARS-CoV-2 S immunoassay (Roche Diagnostics GmbH, Basel, Switzerland) at Toyama University Hospital. At 2wA3D, 3mA3D, and 6mA3D, the upper limit of quantification was 25,000.0 U/mL, and measurements of >25,000.0 U/mL were considered as 25,000.0 U/mL for further statistical calculations. However, at 2wA4D, most serum samples exceeded the upper limit of quantification; therefore, these samples were diluted 4-fold manually before measurement. Measurements of >100,000.0 U/mL at 2wA4D were considered as 100,000.0 U/mL for further statistical calculations. The serum concentration of anti-SARS-CoV-2 nucleocapsid antibodies was measured using an Elecsys anti-SARS-CoV-2 immunoassay (Roche Diagnostics GmbH). The anti-nucleocapsid antibody levels were expressed as a cutoff index value; values of ≥1.0 were considered positive for anti-nucleocapsid antibodies.

**Vaccine-related symptoms after the fourth second booster dose of vaccination.** Data on adverse effects after the second booster dose (mRNA-1273 or BNT162b2 bivalent vaccine) were obtained using the same questionnaire when blood samples were collected at 2wA4D, as previously described (25). Data on participants' age and sex were extracted from the database from the previous study (25). Items on the following adverse effects postvaccination were included: local (pain at the injection site, redness, swelling, hardness, local muscle pain, feeling of warmth, itching, and others) and systemic (fever [temperature, ≥37.5°C], general fatigue, headache, nasal discharge, abdominal pain, nausea, diarrhea, myalgia, joint pain, swelling of the lips and face, hives, cough, and others) symptoms.

**Statistical analysis.** Statistical analysis was performed using the Mann-Whitney test to compare nonparametric groups. Friedman's test with Dunn's test was used for multiple comparisons among the

three paired groups. Correlations between the test findings were expressed using Pearson's correlation coefficient. Data were analyzed using Prism version 9.4.1 (GraphPad Software, San Diego, CA). Statistical significance between different groups was defined as $P < 0.05$. Data are expressed as the median with interquartile range (IQR).

**Ethics approval.** This study was performed in accordance with the Declaration of Helsinki and approved by the ethical review board of the University of Toyama (approval no. R2019167). Written informed consent was obtained from all participants.

**Data availability.** All data are provided in the manuscript and supplementary information.

## SUPPLEMENTAL MATERIAL

Supplemental material is available online only.
**SUPPLEMENTAL FILE 1**, PDF file, 0.6 MB.

## ACKNOWLEDGMENTS

We thank the staff at Toyama University Hospital for their help in collecting the specimens and questionnaires.

Conceptualization: H.K., Y. Morinaga; methodology: H.K., Y. Morinaga, and H.T.; validation: H.K. and Y. Morinaga; formal analysis: H.K. and Y. Morinaga; investigation: H.K., Y. Morinaga, H.Y., Y. Yoshida, M.K., Y. Murai, K.K. (neutralizing assay), and H.N. (commercial antibody test); resources: H.T., Y.S. (generating pseudotyped viruses), C.O., Y. Matsuura (generating plasmids), H.K., Y.T., M.K., Y. Murai, K.K., A.U., Y. Miyajima, and K.N. (serum sample collection); data curation: H.K. and Y. Morinaga; writing-original draft preparation: H.K., Y. Morinaga, and H.T.; writing-review and editing: H.K., Y. Morinaga, and H.T.; visualization: H.K. and Y. Morinaga; supervision: Y. Matsuura, H.N., and Y. Yamamoto; project administration: Y. Morinaga and Y. Yamamoto; funding acquisition: H.K., Y. Morinaga, H.T., H.N., and Y. Yamamoto.

We have no conflicts of interest to declare.

This study was supported by the Research Program on Emerging and Re-emerging Infectious Diseases from the AMED (grant no. JP20he0622035) (Y. Morinaga, H.T., and Y. Yamamoto) and (grant no. JP21fk0108588) (Y. Morinaga and H.T.), a research funding grant from the president of the University of Toyama (Y.M., H.N., and Y. Yamamoto), Toyama Pharmaceutical Valley Development Consortium (Y.M., H.N., and Y. Yamamoto), Morinomiyako Medical Research Foundation (H.K.), Kurozumi Medical Foundation (H.K.), The Hokuriku Bank grant-in-aid for Young Scientists (H.K.), and Japan Society for the Promotion of Science (JSPS) KAKENHI grant no. JP22K20768 (H.K.). The funding bodies played no role in the study design, collection, analysis, or interpretation of data, or in writing the manuscript.

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
