## [Reviewer comments · Microbiology Spectrum]

Microbiology Spectrum

Neutralizing antibody response of the wild-type/Omicron BA.1 bivalent vaccine as the second booster dose against Omicron BA.2 and BA.5

Hitoshi Kawasuji, Yoshitomo Morinaga, Hideki Tani, Yumiko Saga, Hiroshi Yamada, Yoshihiro Yoshida, Yusuke Takegoshi, Makito Kaneda, Yushi Murai, Kou Kimoto, Akitoshi Ueno, Yuki Miyajima, Kentaro Nagaoka, Chikako Ono, Yoshiharu Matsuura, HIDEKI NIIMI, and Yoshihiro Yamamoto

Corresponding Author(s): Yoshitomo Morinaga, Toyama Daigaku

Review Timeline:

Submission Date:	December 14, 2022
Editorial Decision:	January 23, 2023
Revision Received:	February 10, 2023
Accepted:	March 4, 2023

Editor: Day-Yu Chao

Reviewer(s): Disclosure of reviewer identity is with reference to reviewer comments included in decision letter(s). The following individuals involved in review of your submission have agreed to reveal their identity: Pragya D. Yadav (Reviewer #1)

Transaction Report:

DOI: <https://doi.org/10.1128/spectrum.05131-22>

January 23, 2023

Dr. Yoshitomo Morinaga
Toyama Daigaku
Department of Microbiology
2630 Sugitani
Toyama 930-0194
Japan

Re: Spectrum05131-22 (Efficacy of the wild-type/Omicron BA.1 bivalent vaccine as the second booster dose against Omicron BA.2 and BA.5)

Dear Dr. Yoshitomo Morinaga:

Link Not Available

Sincerely,

Day-Yu Chao

Journals Department
Reviewer comments:

Reviewer #1 (Comments for the Author):

Line 230: Please specify the duration instead of just mentioning few months.

Line 240-241: Please mention the percentage of participants instead of mentioning many participants.

Line 269: The authors have mentioned - were not randomly divided in two groups. Please rephrase the sentence.

Line 95-96: Reference for the statement needs to be provided.

Line 112; The authors have mentioned that the not much is known about the clinical efficacy of the bivalent vaccines. In this study also the authors have not assessed the clinical efficacy of the bivalent vaccine. The statement needs to be rephrased.

One of the other major limitation of the study is no data regarding the number of participants who became infected after receiving the second booster dose with the prevailing Omicron variants at that timepoints.

Reviewer #2 (Comments for the Author):

The manuscript entitled "Efficacy of the wild-type/Omicron BA.1 bivalent vaccine as the second booster dose against Omicron BA.2 and BA.5" submitted to Microbiology Spectrum by Dr. Yoshitomo Morinaga, presents real world data regarding the immunogenicity and adverse events of a second booster (4th dose) with the Pfizer Omicron BA.1 bivalent vaccine vs. Moderna WT vax. The authors report a higher elicitation of anti-omicron (BA.1, BA.2 and BA.5) neutralizing antibodies with the bivalent vaccine compared to the monovalent approach.

The authors describe a couple of major limitation related to de present study (i.e., the small number of individuals receiving the bivalent vaccine, thee age range, the absence of background information as for example comorbidities, etc.).

This is a prospective study and the continuation of previously published data: Effectiveness of the third dose of BNT162b2 vaccine on neutralizing Omicron variant in the Japanese population. J Infect Chemother 28:1273-8. The present study adds novel and important information as it is the immunogenicity of bivalent vaccine, a longer follow-up period after previous boost, etc. This manuscript is overall, well written and methodologically correct. The English could be improved in order to make it more readable, though.

One major observation I have is that the authors have misinterpreted the data at the time they make the main conclusion, since VAX efficacy was not measured in the present study; it is all about immunogenicity. This overstatement has to be corrected all along the manuscript including the title.

Line 1: "Efficacy of the wild-type/Omicron BA.1 bivalent vaccine as the second booster dose against Omicron BA.2 and BA.5"

Line 60: "However, there is a lack of real-world data on the effectiveness of bivalent vaccines as second boosters on the dominant Omicron sublineages, including BA.2 and BA.5" this is true as introductory statement, but then DO NOT mix it with your conclusion, since not efficacy was assessed here. Again in line 111: "These bivalent vaccines have already been used, but not much information is available on their clinical efficacy."

Here in line 113- 115, you postulate both concepts in one sentence: "It is crucial to assess newly authorized vaccines for different communities and provide an accurate understanding of their efficacy and side effects. We conducted a prospective longitudinal study to assess the safety, immunogenicity, and reactogenicity ...". Lack of data on efficacy is true, but you did nothing to improve that in the present study, so focus on what you are solving assessing here.

The first sentence of the discussion is closer to what I think is correct statement for this study: Line 203-204: "Although Omicron BA.1-containing bivalent vaccines have been authorized,real-world data validating their safety and antibody responses remain scarce."

Other limitations of the present work are the absence of a later measurement after the 2nd booster and the lack of testing NEUT activity against novel Omicron circulating variants (i.e., XBB.1.5, CH.1.1., CA.3.1., etc). These could be mentioned in the discussion.

Related to this last subject: discussion could be enriched with the incorporation of recent studies exploring the immunogenicity of monovalent vs bivalent boosting approaches. One example for this is the recent preprint by Shan-Lu Liu "Extraordinary evasion of Nab response by XBB.1.5, CH.1.1., CA.3.1 variants" doi: <https://doi.org/10.1101/2023.01.16.524244>. There are other references, including peer-reviewed ones, about this particular subject.

In the abstract: Line 68-70: "Although the neutralization activity of the pooled sera of the WT+BA.1 group was the lowest against BA.5, the activities against BA.2 and BA.5 were higher than those of the WT group in both pseudotyped and live virus assays." This sentence is confusing and I do not observe that neutralization activity of the pooled sera of the WT+BA.1 group was the lowest against BA.5 (FIGURE 3).

Line 81-82: "... the bivalent (WT+BA.1) vaccine elicited higher neutralization against Omicron BA.2 and BA.5 subvariant", should be elicited higher levels/titers of neutralizing antibodies against...

Staff Comments:

Preparing Revision Guidelines

Please return the manuscript within 60 days; if you cannot complete the modification within this time period, please contact me. If you do not wish to modify the manuscript and prefer to submit it to another journal, please notify me of your decision immediately so that the manuscript may be formally withdrawn from consideration by Microbiology Spectrum.

Responses to reviewers' comments

We thank the reviewers for their incisive comments and suggestions. Below are our responses in red. The revised manuscript and our responses were checked for grammar and edited by professional English editing service before re-submission.

Reviewer 1

Line 230: Please specify the duration instead of just mentioning few months.

Response: We agree with the reviewer's comment. We specified the duration and revised the following sentence as suggested:

“In fact, vaccine effectiveness **against infection** waned remarkably within **three** months of administration in clinical settings (19, 20).” (Line 239–241)

Line 240-241: Please mention the percentage of participants instead of mentioning many participants.

Response: Thank you for your valuable suggestion. We specified the percentage and revised the following sentence as suggested:

“Furthermore, **over 66.7% of** participants who experienced symptoms after the first booster dose showed the same symptoms after the second booster dose.” (Line 250–251)

Line 269: The authors have mentioned - were not randomly divided in two groups. Please rephrase the sentence.

Response: We are grateful for the valuable comments. We rephrased and added the following sentences to explain why only a small group of individuals received the bivalent vaccine.

“In the present study, participants who provided blood samples at least once after the third dose were eligible for analysis and were **voluntarily** divided into two groups: those who received the original mRNA-1273 vaccine and those who received the recently authorized Pfizer BNT162b2 bivalent Omicron BA.1-containing vaccine as the second booster (fourth dose). (Lines 283–287)

“The second booster dose had been promoted in our hospital according to the national vaccine campaigns before the bivalent vaccine was authorized. Therefore, most

healthcare workers (HCWs) had already received the second booster dose of the original monovalent vaccine, while those who had waited for the bivalent vaccines received the bivalent vaccine as a second booster dose later.” (Lines 289–294)

Line 95-96: Reference for the statement needs to be provided.

Response: We agree with the reviewer’s comment. We revised the following sentence and added the references, as suggested.

“Vaccination programs against COVID-19 have been conducted worldwide; however, vaccine coverage varies per country based on the social circumstances of each country (5, 6).” (Lines 95–97)

5. Mathieu E, Ritchie H, Ortiz-Ospina E, Roser M, Hasell J, Appel C, Giattino C, Rodés-Guirao L. 2021. A global database of COVID-19 vaccinations. *Nature Human Behaviour* 5:947-953.
6. Wouters OJ, Shadlen KC, Salcher-Konrad M, Pollard AJ, Larson HJ, Teerawattananon Y, Jit M. 2021. Challenges in ensuring global access to COVID-19 vaccines: production, affordability, allocation, and deployment. *The Lancet* 397:1023-1034.

Line 112; The authors have mentioned that the not much is known about the clinical efficacy of the bivalent vaccines. In this study also the authors have not assessed the clinical efficacy of the bivalent vaccine. The statement needs to be rephrased.

Response: Thank you for your valuable suggestions. We rephrased the following sentence as suggested:

“These bivalent vaccines have already been used, but not much information is available on their immunogenicity and safety compared to the monovalent vaccines.” (Lines 109–111)

One of the other major limitation of the study is no data regarding the number of participants who became infected after receiving the second booster dose with the prevailing Omicron variants at that timepoints.

Response: We are grateful for the valuable comment. As you and reviewer 2 thoughtfully and insightfully suggested, the vaccine efficacy was not assessed in the

present study. Because the follow-up time after the second booster dose was limited in the present study, we should conduct a follow-up study to monitor the incidences of symptomatic and asymptomatic SARS-CoV-2 infections in both groups. We added the following sentences as one of the major limitations.

“Fourth, the present study did not assess the vaccine efficacy and investigate the number of participants who infected after receiving the second booster dose because the follow-up time after the second booster was limited. Future studies should focus on vaccine effectiveness and longitudinally monitor the incidences of symptomatic and asymptomatic SARS-CoV-2 infections in the WT and WT+BA.1 groups.” (Lines 265–269)

Reviewer 2

The manuscript entitled "Efficacy of the wild-type/Omicron BA.1 bivalent vaccine as the second booster dose against Omicron BA.2 and BA.5" submitted to Microbiology Spectrum by Dr. Yoshitomo Morinaga, presents real world data regarding the immunogenicity and adverse events of a second booster (4th dose) with the Pfizer Omicron BA.1 bivalent vaccine vs. Moderna WT vax. The authors report a higher elicitation of anti-omicron (BA.1, BA.2 and BA.5) neutralizing antibodies with the bivalent vaccine compared to the monovalent approach.

The authors describe a couple of major limitation related to de present study (i.e., the small number of individuals receiving the bivalent vaccine, thee age range, the absence of background information as for example comorbidities, etc.).

This is a prospective study and the continuation of previously published data: Effectiveness of the third dose of BNT162b2 vaccine on neutralizing Omicron variant in the Japanese population. J Infect Chemother 28:1273-8. The present study adds novel and important information as it is the immunogenicity of bivalent vaccine, a longer follow-up period after previous boost, etc. This manuscript is overall, well written and methodologically correct. The English could be improved in order to make it more readable, though.

One major observation I have is that the authors have misinterpreted the data at the time they make the main conclusion, since VAX efficacy was not measured in the present study; it is all about immunogenicity. This overstatement has to be corrected all along the manuscript including the title.

Response: Thank you for your valuable suggestions, and we sincerely apologize for our misleading conclusion. We rephased it throughout the manuscript as follows:

Line 1: "Efficacy of the wild-type/Omicron BA.1 bivalent vaccine as the second booster dose against Omicron BA.2 and BA.5"

Response: We are grateful for the valuable comments. We revised the title to "Neutralizing antibody response of the wild-type/Omicron BA.1 bivalent vaccine as the second booster dose against Omicron BA.2 and BA.5" (Lines 1–2)

Line 60: "However, there is a lack of real-world data on the effectiveness of bivalent vaccines as second boosters on the dominant Omicron sublineages, including BA.2 and BA.5" this is true as introductory statement, but then DO NOT mix it with your conclusion, since not efficacy was assessed here.

Response: We agree with the reviewer's comment. We revised the following sentence as suggested:

"However, there is a lack of real-world data on the immunogenicity of bivalent vaccines as second boosters against the dominant Omicron sublineages, including BA.2 and BA.5." (Line 59–61)

Again in line 111: "These bivalent vaccines have already been used, but not much information is available on their clinical efficacy."

Response: Thank you for your suggestion. We revised the following sentence:

"These bivalent vaccines have already been used, but not much information is available on their immunogenicity and safety compared to the monovalent vaccines." (Lines 109–111)

Here in line 113- 115, you postulate both concepts in one sentence: "It is crucial to assess newly authorized vaccines for different communities and provide an accurate understanding of their efficacy and side effects. We conducted a prospective longitudinal study to assess the safety, immunogenicity, and reactogenicity ...". Lack of data on efficacy is true, but you did nothing to improve that in the present study, so focus on what you are solving assessing here.

Response: We are grateful for the valuable comments. We revised the following sentence to focus on what was assessed and evaluated in the present study.

"It is crucial to assess newly authorized vaccines in real-world clinical settings and

provide an accurate understanding of their humoral immune response to circulating variants and side effects.” (Lines 112–114)

The first sentence of the discussion is closer to what I think is correct statement for this study: Line 203-204: "Although Omicron BA.1-containing bivalent vaccines have been authorized, real-world data validating their safety and antibody responses remain scarce."

Response: According to your thoughtful comments, we revised the following sentences and added the reference (14) in the Discussion section:

“This clinical study showed that a second booster dose with the Omicron BA.1 bivalent vaccine elicited higher levels of neutralizing antibodies against BA.1, BA.4, BA.5, and BA.2.75 than the original monovalent vaccine (14). However, data are still insufficient to ensure the safety and immunogenicity of the bivalent Omicron-containing vaccines. It is thus difficult for clinicians and policymakers to decide which booster to recommend while considering a delay of vaccination or vaccine shortages.” (Lines 204–210)

14. Winokur P, Gayed J, Fitz-Patrick D, Thomas SJ, Diya O, Lockhart S, Xu X, Zhang Y, Bangad V, Schwartz HI, Denham D, Cardona JF, Usdan L, Ginis J, Mensa FJ, Zou J, Xie X, Shi P-Y, Lu C, Buitrago S, Scully IL, Cooper D, Koury K, Jansen KU, Türeci Ö, Şahin U, Swanson KA, Gruber WC, Kitchin N. 2023. Bivalent Omicron BA.1-adapted BNT162b2 booster in adults older than 55 years. *New England Journal of Medicine* 388:214-227.

We also revised the following sentence in the Discussion section:

“Similar to our previous studies about the immunogenicity and safety following primary and booster vaccinations (25, 29), a positive relationship was observed between higher antibody responses and some specific adverse reactions.” (Lines 247–250)

Other limitations of the present work are the absence of a later measurement after the 2nd booster and the lack of testing NEUT activity against novel Omicron circulating variants (i.e., XBB.1.5, CH.1.1., CA.3.1., etc). These could be mentioned in the discussion.

Related to this last subject: discussion could be enriched with the incorporation of recent studies exploring the immunogenicity of monovalent vs bivalent boosting

approaches. One example for this is the recent preprint by Shan-Lu Liu "Extraordinary evasion of Nab response by XBB.1.5, CH.1.1., CA.3.1 variants" doi: <https://doi.org/10.1101/2023.01.16.524244>. There are other references, including peer-reviewed ones, about this particular subject.

Response: Thank you for your valuable suggestions. We added the following sentences to incorporate the recent studies exploring the immunogenicity of the monovalent and bivalent vaccines against novel Omicron circulating variants in the Discussion section. In addition, we added the absence of a later measurement after the second booster dose and the lack of testing neutralization activity against these newly emerged BQ.1.1, XBB.1, XBB.1.5, CH.1.1, and CA.3.1 as one of the major limitations.

“Following the approval of Omicron BA.1-containing bivalent vaccines, the current available Omicron BA.4/BA.5-adapted bivalent vaccines were rapidly authorized and are now widely used (12); however, the virus continues to evolve, and the BA.4 and BA.5 variants are no longer dominant in several countries (9). The convergent evolution of BA.2, BA.4, and BA.5 lineages has led to the emergence of several new subvariants, including BQ.1.1, XBB.1.5, CH.1.1, and CA.3.1 (9, 20). Previous studies reported that a first or second booster dose of the bivalent BA.4/BA.5 vaccine was more immunogenic than the original monovalent vaccine against BQ.1.1, XBB, and XBB.1 (21, 22). However, recent studies have consistently demonstrated that the bivalent booster vaccines did not produce a robust neutralizing antibody response against BQ.1.1, XBB.1, XBB.1.5, CH.1.1, and CA.3.1 (20, 23). The newly emerged sublineages have accumulated additional spike mutations that enhanced immune evasion (24). It is, therefore, critical to monitor viral evolution and the impact on the immunogenicity and efficacy of the bivalent vaccines.” (Lines 222–234)

“Third, the long-term persistence of neutralizing antibody levels after the second booster dose was not assessed, and neutralization activity against novel Omicron circulating variants (BQ.1.1, XBB.1.5, CH.1.1, and CA.3.1) were not measured.” (Lines 262–269)

20. Qu P, Faraone JN, Evans JP, Zheng Y-M, Carlin C, Anghelina M, Stevens P, Fernandez S, Jones D, Panchal A, Saif LJ, Oltz EM, Xu K, Gumina RJ, Liu S-L. 2023. Extraordinary evasion of neutralizing antibody response by Omicron XBB.1.5, CH.1.1 and CA.3.1 variants. bioRxiv doi:10.1101/2023.01.16.524244:2023.01.16.524244.

21. Zou J, Kurhade C, Patel S, Kitchin N, Tompkins K, Cutler M, Cooper D, Yang Q, Cai H, Muik A, Zhang Y, Lee D-Y, Sahin U, Anderson AS, Gruber WC, Xie X, Swanson KA, Shi P-Y. 2022. Improved neutralization of Omicron BA.4/5, BA.4.6, BA.2.75.2, BQ.1.1, and XBB.1 with bivalent BA.4/5 vaccine. *bioRxiv* doi:10.1101/2022.11.17.516898:2022.11.17.516898.
22. Davis-Gardner ME, Lai L, Wali B, Samaha H, Solis D, Lee M, Porter-Morrison A, Hentenaar IT, Yamamoto F, Godbole S, Liu Y, Douek DC, Lee FE-H, Roupheal N, Moreno A, Pinsky BA, Suthar MS. 2022. Neutralization against BA.2.75.2, BQ.1.1, and XBB from mRNA Bivalent Booster. *New England Journal of Medicine* 388:183-185.
23. Kurhade C, Zou J, Xia H, Liu M, Chang HC, Ren P, Xie X, Shi PY. 2022. Low neutralization of SARS-CoV-2 Omicron BA.2.75.2, BQ.1.1 and XBB.1 by parental mRNA vaccine or a BA.5 bivalent booster. *Nature Medicine* doi:10.1038/s41591-022-02162-x.
24. Carabelli AM, Peacock TP, Thorne LG, Harvey WT, Hughes J, de Silva TI, Peacock SJ, Barclay WS, de Silva TI, Towers GJ, Robertson DL, Consortium C-GU. 2023. SARS-CoV-2 variant biology: immune escape, transmission and fitness. *Nature Reviews Microbiology* doi:10.1038/s41579-022-00841-7.

In the abstract: Line 68-70: "Although the neutralization activity of the pooled sera of the WT+BA.1 group was the lowest against BA.5, the activities against BA.2 and BA.5 were higher than those of the WT group in both pseudotyped and live virus assays." This sentence is confusing, and I do not observe that neutralization activity of the pooled sera of the WT+BA.1 group was the lowest against BA.5 (FIGURE 3).

Response: We are grateful for the valuable comments. To avoid confusion, we revised the following sentence as suggested:

"Although the neutralization activities of the pooled sera were lower against BA.5 compared to other variants in both groups, the activities against BA.2 and BA.5 of the WT+BA.1 group were higher than those of the WT group in both pseudotyped and live virus assays." (Lines 68–70)

Line 81-82: "... the bivalent (WT+BA.1) vaccine elicited higher neutralization against Omicron BA.2 and BA.5 subvariant", should be elicited higher levels/titers of neutralizing antibodies against...

Response: Thank you for your suggestions. We revised the following sentence as

suggested.

“Compared with the original monovalent vaccine, the bivalent (WT+BA.1) vaccine elicited higher levels of neutralizing antibodies against the Omicron BA.2 and BA.5 subvariants.” (Lines 80–82)

In addition, we also revised the following sentence in the Discussion section.

“Similar to previous studies, we showed that the bivalent (WT+BA.1) vaccine elicited higher levels of neutralizing antibodies against Omicron BA.2 and BA.5 subvariants than the monovalent vaccine (11).” (Lines 213–215)

February 27, 2023

Dr. Yoshitomo Morinaga
Toyama Daigaku
Department of Microbiology
2630 Sugitani
Toyama 930-0194
Japan

Re: Spectrum05131-22R1 (Neutralizing antibody response of the wild-type/Omicron BA.1 bivalent vaccine as the second booster dose against Omicron BA.2 and BA.5)

Dear Dr. Yoshitomo Morinaga:

Your manuscript has been accepted, and I am forwarding it to the ASM Journals Department for publication. You will be notified when your proofs are ready to be viewed.

Sincerely,

Day-Yu Chao
Editor, Microbiology Spectrum
